# Machine Learning-Based Integration of High-Resolution Wildfire Smoke Simulations and Observations for Regional Health Impact Assessment

**DOI:** 10.3390/ijerph16122137

**Published:** 2019-06-17

**Authors:** Yufei Zou, Susan M. O’Neill, Narasimhan K. Larkin, Ernesto C. Alvarado, Robert Solomon, Clifford Mass, Yang Liu, M. Talat Odman, Huizhong Shen

**Affiliations:** 1School of Environmental and Forest Sciences, University of Washington, Seattle, WA 98195, USA; alvarado@uw.edu (E.C.A.); robert.airfire@gmail.com (R.S.); 2Pacific Wildland Fire Sciences Laboratory, U.S. Forest Service, Seattle, WA 98103, USA; smoneill@fs.fed.us (S.M.O.); larkin@fs.fed.us (N.K.L.); 3Department of Atmospheric Sciences, University of Washington, Seattle, WA 98195, USA; cmass@uw.edu; 4Rollins School of Public Health, Emory University, Atlanta, GA 30322, USA; yang.liu@emory.edu; 5School of Civil and Environmental Engineering, Georgia Institute of Technology, Atlanta, GA 30332, USA; talat.odman@ce.gatech.edu (M.T.O.); huizhong.shen@ce.gatech.edu (H.S.)

**Keywords:** fire smoke modeling, PM_2.5_ air pollution, machine learning-based data fusion, health impact assessment

## Abstract

Large wildfires are an increasing threat to the western U.S. In the 2017 fire season, extensive wildfires occurred across the Pacific Northwest (PNW). To evaluate public health impacts of wildfire smoke, we integrated numerical simulations and observations for regional fire events during August-September of 2017. A one-way coupled Weather Research and Forecasting and Community Multiscale Air Quality modeling system was used to simulate fire smoke transport and dispersion. To reduce modeling bias in fine particulate matter (PM_2.5_) and to optimize smoke exposure estimates, we integrated modeling results with the high-resolution Multi-Angle Implementation of Atmospheric Correction satellite aerosol optical depth and the U.S. Environmental Protection Agency AirNow ground-level monitoring PM_2.5_ concentrations. Three machine learning-based data fusion algorithms were applied: An ordinary multi-linear regression method, a generalized boosting method, and a random forest (RF) method. 10-Fold cross-validation found improved surface PM_2.5_ estimation after data integration and bias correction, especially with the RF method. Lastly, to assess transient health effects of fire smoke, we applied the optimized high-resolution PM_2.5_ exposure estimate in a short-term exposure-response function. Total estimated regional mortality attributable to PM_2.5_ exposure during the smoke episode was 183 (95% confidence interval: 0, 432), with 85% of the PM_2.5_ pollution and 95% of the consequent multiple-cause mortality contributed by fire emissions. This application demonstrates both the profound health impacts of fire smoke over the PNW and the need for a high-performance fire smoke forecasting and reanalysis system to reduce public health risks of smoke hazards in fire-prone regions.

## 1. Introduction

Toxic smoke from fire hotspots during the fire season poses a serious health threat in many fire-prone regions around the world. During 1997–2006, global annual average mortality attributable to landscape fire smoke was estimated at 339,000 (interquartile range: 226,000–600,000), with the most affected regions including sub-Sahara Africa (157,000) and Southeast Asia (110,000) [1]. In a similar study focusing on the continental USA for 2008–2012, researchers assessed not only health impacts regarding excess mortality and morbidity but also the economic value of these events [2]. Estimated annual health consequences of exposure to wildland fire smoke during that period included 5200–8500 respiratory hospital admissions, 1500–2500 cardiovascular hospital admissions, 1500–2500 premature deaths related to short-term exposure to fine particulate matter (PM_2.5_), and 8700–32,000 premature deaths related to long-term exposure to PM_2.5_. For the five-year period, the net present economic costs in 2010 dollars were estimated at $63B from short-term exposure to PM_2.5_ and $450B from long-term exposure [2]. The USA regions most affected by fire smoke include northern California, Oregon, and Idaho in the West, and Florida, Louisiana, and Georgia in the East [2]. These major fire-prone regions face different types of fire threat, with large contributions from both wildfires and prescribed fires in the western USA, and a dominant role of agricultural burning and prescribed fires in the southeastern USA [3]. Recent reviews have assessed the health effects from wildfire smoke exposure [4], summarized heterogeneous respiratory health effects of fire smoke in different demographic subgroups [5], and compared community smoke exposure from wildfires and prescribed fires between the western and eastern USA [6]. These studies all highlight the need to better understand both short-term and long-term fire smoke consequences on subgroups of the population [7]. 

To evaluate the USA population risk from fire-originated PM_2.5_ pollution exposure, Rappold et al. [8] developed a Community Health-Vulnerability Index (CHVI) based on key socioeconomic factors, including county prevalence rates for multiple respiratory and cardiovascular diseases, population age groups, and socioeconomic status indicators. Results showed that 10% of the USA population (30.5 million) lived in areas with high ambient PM_2.5_ attributable to fire emissions, and 10.3 million people had experienced unhealthy air quality levels for more than 10 days because of fire smoke pollution. As expected, most of these people under threat of fires and smoke lived close to fire-prone regions, especially in the western USA where high PM_2.5_ concentrations were collocated with a dense distribution of large wildfires.

Given its unique landscape and ecosystems including extensive forest coverage, the Pacific Northwest (PNW) is among these regions heavily affected by smoke during the fire season from late spring to late autumn [9]. In contrast to most other areas of the USA, that have continuously improved air quality during the last three decades, the fire-prone Northwestern region shows increasing trends in both ground-based PM_2.5_ pollution extremes and space-based aerosol optical depth (AOD) [10]. These increasing pollution trends have been attributed to a prevalence of wildfires across the Northwest [10], as supported by many other regional smoke monitoring and modeling studies. In the fire seasons of 2005–2008, Strand et al. [11] conducted field measurements during different types of fire events in the PNW and found significant impacts of fire smoke on local and regional air quality, with good correlations between daily active burning areas and measured PM_2.5_ concentrations. To quantify fire impacts on regional air pollution, researchers used a regional air quality modeling system, Air Indicator Report for Public Awareness and Community Tracking, AIRPACT-3 (http://lar.wsu.edu/airpact/), and a fire emission modeling framework, BlueSky [12], to simulate O_3_, PM_2.5_, and tropospheric NO_2_ across the PNW [13,14]. Although the AIRPACT system overestimated regional tropospheric NO_2_ during fire events [13], it well captured fire impacts on surface PM_2.5_ in this region [14]. It’s noted that some important atmospheric processes such as secondary aerosol formation and transformation by complex homogenous and heterogenous reactions in fire smoke plumes are poorly represented in current chemical transport models (CTMs). Many recent studies have investigated chemical reactions of biomass burning products like catechol and phenolic compounds through oxidation at the air-water interface [15,16,17] or photodegradation in the aqueous phase [18,19,20]. These complex chemical pathways would affect secondary organic aerosol (SOA) formation and transformation in fire plumes and even removal of these biomass burning products [21]. Though here we used a traditional organic aerosol treatment (CMAQ-AE6) that has limited modeling capability to reproduce these complex reactions, we want to point out that continuous model development [22,23] is ongoing to improve the aerosol modeling capability for a better understanding of the role of aerosols in the climate system and human health. 

However, some obstacles remain when applying these observation- and model-based results in assessing the health consequences of regional fire smoke exposure. Source-specific health assessment studies are usually hindered by both the lack of high-quality fire smoke exposure estimates and the uncertainty of exposure-response relationships [24]. Most current smoke exposure estimates are based on stationary and temporary air pollutant monitors as well as satellite remote sensing data [25]; these have limitations in spatiotemporal coverage, data quality of surface pollution levels, and quantification of fire smoke contributions. In comparison, chemical transport models have unique advantages in addressing these problems with smoke exposure and assessment of resulting health impacts, because CTMs consider both fire and non-fire emissions. For example, Youssouf et al. [26] reviewed and compared available fire exposure assessment approaches, including self-reported questionnaires, ground measurements, satellite retrievals, and CTMs. They also used a hybrid model based on these approaches to estimate a country-level wildfire emission inventory during 2006–2010 in Europe. However, limited data elsewhere restricts the application of such complex hybrid models in many regions, including the USA.

In this study, we applied a machine learning (ML)-based data integration approach based on the three major assessment elements—ground monitoring PM_2.5_ concentrations, satellite AOD retrievals, and source-oriented CTM simulations—to identify fire source contributions to regional PM_2.5_ pollution and population health exposure. We used this integrated assessment of smoke concentrations to conduct a case study for a series of large fire events over the PNW region during summer 2017, evaluating PM_2.5_ modeling performance as well as regional health effects of the wildfire smoke by separating fire emission contributions from other sources. 

## 2. Data Materials and Modeling Methods 

We incorporated a coupled regional air quality modeling system based on the Weather Research and Forecasting (WRF) model version 3.7 [27] and the Community Multiscale Air Quality (CMAQ) model version 5.2 [28] into the BlueSky fire smoke modeling framework [12]. The WRF model (https://www.mmm.ucar.edu/weather-research-and-forecasting-model) is a mesoscale numerical weather prediction system designed for both atmospheric research and operational forecasting applications. The CMAQ model (https://www.epa.gov/cmaq) is a numerical air quality model developed to simulate atmospheric chemistry and dynamic transport of air pollutants. We used a one-way coupled WRF-CMAQ modeling system with the CMAQ simulation driven by WRF meteorological outputs without consideration of chemistry feedbacks to the weather. Although previous studies [29,30] suggested significant effects of two-way interactions between chemistry and meteorology through radiative and cloud processes, such feedback effects are beyond the scope of this work, given the expensive computational burden and large uncertainties in these feedback processes. Here, the data integration methods will be introduced specifically to reduce modeling bias related to such limitations of the modeling framework and simulation settings. Fire emissions for numerical simulations were estimated based on the BlueSky framework [12], which incorporates satellite-based fire detection information (e.g., fire size and location) with fuel loadings and moisture conditions to calculate fuel consumption and fire emissions. All fire and non-fire emissions were processed by the Sparse Matrix Operator Kernel Emissions (SMOKE) model [31] in the CMAQ system [28] to generate model-ready gridded emission inputs after spatial (i.e., horizontal and vertical) and temporal allocation as well as chemical speciation.

To isolate fire source contributions to the regional air pollution episode and consequent public health effects, we designed two WRF-CMAQ simulation experiments: A control experiment (CTRL) with only non-fire emission sources (i.e., anthropogenic and natural sources) and a sensitivity experiment (SENS) with both fire and non-fire sources (Table 1). The SENS experiment with all sources was compared with space- and ground-based observations for model evaluation. 

Uncertainties in model inputs and physical assumptions may result in large bias in model outputs such as PM_2.5_ concentrations. Therefore, we applied several ML-based data fusion algorithms to reduce these modeling biases, rather than directly using modeling results for assessing regional health consequences of fire smoke exposure. Specifically, we designed a two-step approach based on three ML algorithms: Ordinary multi-linear regression (MLR), a generalized boosting model (GBM), and a random forest (RF) method. The last two algorithms were suggested by a previous study [33], in which the researchers compared a set of 11 ML algorithms and demonstrated the favorable performance of the GBM and RF methods for predicting PM_2.5_ concentrations during a 2008 fire event in California. Specifically, a two-step data integration approach was followed in this work:(1)Step 1: Gap filling for spatiotemporal missing values in the Multi-Angle Implementation of Atmospheric Correction (MAIAC) satellite AOD retrievals [34]. This was based on (1) three hourly meteorological variables including total cloud fraction, cloud liquid water content, and surface water vapor mixing ratio from the WRF outputs, (2) two geographical variables including terrain elevation and vegetation coverage, and (3) the simulated hourly AOD from the WRF-CMAQ SENS experiment;(2)Step 2: Data fusion for optimizing daily surface PM_2.5_ concentrations from the WRF-CMAQ SENS experiment, based on (1) daily averages of observational AirNow surface PM_2.5_ measurements, (2) gap-filled AOD from Step 1, and (3) six meteorological variables, including surface wind speed and directions at 10 m, surface air temperature at 2 m, relative humidity at 2 m, precipitation rates at surface on a log scale, and planetary boundary layer heights from the North American Regional Reanalysis (NARR) data [35] produced by the National Centers for Environmental Prediction.

We used the MLR algorithm as the benchmark for the other two advanced ML algorithms (GBM and RF) and compared the performance of the three statistical algorithms in Step 1 and 2. An MLR model can be described by the following equation: (1)$$y=\alpha +\sum_{k}\beta_{k}x_{k}+\epsilon ,$$
where x and y are explanatory and dependent variables, α and β are intercept and slope coefficients, ϵ is the error term, and k is the number of explanatory variables (k>1) such as the observational and simulated AOD and meteorological variables. In comparison, the GBM and RF methods are tree-based algorithms using decision trees as base learners. They use multiple features to split the whole training dataset into small subsets like branching of a tree, and then fit models on subset samples for each tree branch. This process is repeated by resampling the original training data with the bootstrap technique so that a large number of trees could grow. They share similar features by combining ensemble model outputs from each individual tree, but they differ on how the trees are built. For instance, GBM is based on shallow trees, with emphasis on poorly predicted instances by previously built models when growing trees sequentially, while RF is based on fully grown trees in parallel by treating all instances equally. RF also uses a random subset of features for each candidate split in the learning process to decorrelate the grown trees. These fundamental differences may affect the modeling performance of GBM and RF in specific cases, but both require parameter selection and model tuning efforts to achieve optimal performance. We used the R packages including “lm” [36], “gbm” [37], and “randomForest” [38] for statistical modeling and data integration.

After data fusion processing, we evaluated modeling performance during the study period (8/15–9/14/2017) from different perspectives including spatial distributions, temporal variations, and aerosol chemical composition. Table 2 lists all the observational datasets used for the model evaluation. 

For aerosol horizontal distributions, we collected the MCD19A2 Version 6 of the Moderate Resolution Imaging Spectroradiometer (MODIS) Terra and Aqua combined MAIAC AOD at 550 nm product [34] from the National Aeronautics and Space Administration (NASA). This relatively new satellite product was successfully used to estimate surface PM_2.5_ concentrations with full-coverage in other regions [40]. For aerosol vertical distributions, we used the L1B v3.00 aerosol cross section product [39] from the Cloud-Aerosol Transport System (CATS) aboard the International Space Station (ISS). Due to the unique orbit path of the ISS, CATS could observe the same spot on Earth at different times each day, providing more comprehensive coverage of the tropics and mid-latitudes than sun-synchronous orbiting satellites [39]. For bulk PM_2.5_ temporal variations, we collected AirNow in situ PM_2.5_ surface concentrations. We also collected in situ aerosol speciation measurements from the Interagency Monitoring of Protected Visual Environments (IMPROVE) program for PM_2.5_ chemical composition. To screen the optimal ML algorithm with the best modeling performance, we applied 10-fold cross-validation (CV) during the model training and parameter tuning processes of the two ML algorithms. The 10-fold CV was conducted by partitioning the whole data samples into 10 equal sized subsets, and then using nine subsets for model training and the remaining subset for testing purpose. The cross-validation process was repeated 10 times (the folds) until each of the subsets was used for validation once. The final model evaluation results were obtained by averaging all the results from the 10 validation processes. We evaluated their performance in terms of multiple statistical metrics such as mean absolute error (MAE), fractional bias (FB), R-squared (R^2^), and root mean squared error (RMSE) (see Appendix A). Lastly, we assessed regional health impacts of fire smoke aerosols based on optimized surface PM_2.5_ estimates and a relative risk function for multiple-cause mortality following the method of a fire smoke-focused health assessment study [1]. Table 3 lists the input data used for the regional health impact assessment. Demographic data were collected from the 2010 USA Census Grids provided by the NASA Socioeconomic Data and Applications Center (SEDAC). Multiple-cause mortality data at the county level over the PNW during and after the fire smoke pollution episode were collected from the 2017 multiple cause of death online database by Wide-ranging ONline Data for Epidemiologic Research (WONDER) of Centers for Disease Control and Prevention (CDC). The multiple-cause mortality attributable to PM_2.5_ exposure during the fire smoke pollution episode was estimated using the following exposure-response function adapted from Johnston et al. [1]:(2)$$Mortality\;attributable\;to\;PM_{2.5}\;exposure=\sum_{PM_{2.5}}^{n}D_{PM_{2.5}}\times M\times \left(RR_{SI}\left(PM_{2.5}\right)-1\right),$$
where $PM_{2.5}$ is daily average surface PM_2.5_ concentrations with minimum and maximum values of 5 and 200 μg m^−3^, respectively. Following Johnston et al. [1], we excluded the grid cells with daily exposure estimates of less than 5 μg m^−3^ and fixed grid cells with exposure estimates larger than 200 μg m^−3^ to a maximum threshold of 200 μg m^−3^. Although there is little evidence for a threshold PM_2.5_ mortality function, the minimum threshold was set to reflect the less certainty of the shape of the exposure-response relationship at low PM_2.5_ levels [41], while the maximum threshold was set to represent the flat shape of the exposure-response relationship at high concentration levels [42]. $D_{PM_{2.5}}$ is the number of days with daily PM_2.5_ at certain levels between each PM_2.5_ concentration interval (i.e., each 1 μg m^−3^ increment between 5 μg m^−3^ and 200 μg m^−3^), n is the total number of concentration intervals, M is the county-level daily average number of multiple cause of deaths between August-December of 2017, and $RR_{SI}$ is a relative risk function for multiple-cause mortality due to short-term PM_2.5_ exposure. We downscaled the county-level multiple cause of deaths (M) to the model-grid scale according to the high-resolution gridded population density data from the 2010 USA Census Grids. Therefore, we were able to estimate population exposure risks with both gridded mortality and PM_2.5_ concentrations at the same resolution of modeling grids (i.e., 4 km). 

## 3. Observational and Modeling Results

### 3.1. The 2017 PNW Fire Smoke Pollution Episode 

The 2017 fire smoke pollution episode in the PNW was characterized by a series of region-wide large wildland fires in the eastern and western PNW mountains, as well as heavy fire smoke from these fires from mid-August to mid-September 2017. Figure 1 shows fire hotspots and smoke over the PNW on 5 September 2017 in the space-based imagery (Figure 1a) detected by the Suomi National Polar-orbiting Partnership (NPP) satellite project (http://www.nasa.gov/NPP), a collaboration between NASA, National Oceanic and Atmospheric Administration (NOAA), and Department of Defense, and the model-based PM_2.5_ simulation (Figure 1b). Major population centers such as the Seattle and Portland metropolitan areas were engulfed by heavy and prolonged smoke during this period. A ground measurement station in Missoula, Montana observed ~500 h heavy smoke episodes, with up to 471 μg m^−3^ of hourly PM_2.5_ [44]. Such prolonged exposure to severe smoke pollution may cause serious public health problems in this region.

Next, we used multiple numerical and statistical modeling techniques to estimate the environmental and health impacts of this fire smoke pollution episode. Similar techniques will also be applied to an operational air quality forecasting system (https://tools.airfire.org) deployed at the Pacific Wildland Fire Sciences Laboratory of the USA Forest Service (USFS) to improve daily air quality forecasting skills as well as to reduce public exposure risks to fire smoke hazards through the interagency Wildland Fire Air Quality Response Program (WFAQRP) led by the USFS. Currently, there are two fire smoke modeling systems in operation at the fire lab, one based on the Hybrid Single Particle Lagrangian Integrated Trajectory (HYSPLIT) model, and the other based on the CMAQ model. The HYSPLIT model was developed to simulate physical dispersion and transport of air pollutants, while the Eulerian-based CMAQ model allows to simulate more comprehensive atmospheric processes such as dynamic advection, diffusion, deposition, and chemical reactions. The operational CMAQ model system also has an advantage to deal with regional leftover fire smoke that is neglected in the HYSPLIT model system. By integrating observations into these simulated leftover concentration fields, the CMAQ system would start from more accurate initial conditions for fire smoke forecast operations in the future. These efforts could improve the quality of fire smoke forecast products and better support public smoke advisories by Air Resource Advisors (ARAs) through the WFAQRP program. 

### 3.2. Model Simulation and Evalution Results

#### 3.2.1. Gap-Filling for MAIAC AOD

We first compared the WRF-CMAQ model-simulated AOD from the SENS experiment with the MAIAC satellite retrievals in terms of fire smoke horizontal distributions. More than half of the PNW is covered by missing values of the MAIAC AOD on 7 September 2017 (Figure 2a); these missing values may be related to regional cloud contamination, mountainous terrain, and high surface albedo issues in the satellite retrieval algorithm [45]. Such missing value problems also exist on other days of the study period and require gap-filling for further application in estimation of surface PM_2.5_. To solve this problem, we implemented ML-based gap-filling processing for missing values in the raw MAIAC AOD products based on WRF-CMAQ simulated AOD and meteorological variables. Although the model simulated AOD are low-biased in general (Figure 2b), they still captured the regional distributions of fire aerosols before and after gap-filling processing with different ML-based algorithms (Figure 2c,d). Both GBM (*r* = 0.96) and RF (*r* = 0.96) methods show greatly improved spatial correlations of gap-filled AOD with satellite retrievals, which increased our confidence of using these techniques in the next step to optimize surface PM_2.5_ concentrations and fire smoke exposures. We also compared the statistical performance in terms of spatial correlation (r) and RMSE values for all modeling days (08/15/2017–09/14/2017) (Appendix A).

Besides horizontal distributions, we also evaluated smoke vertical distributions. Figure 3 shows comparisons of space-based aerosol cross sections retrieved by the CATS system and the WRF-CMAQ model simulation at 11:30 am UTC on September 7, 2017 (Appendix A). The heavy fire smoke layer shows up with a clear vertical structure over the PNW in both satellite and model results. The highest smoke layer reached approximately 10 km above sea level, and the thickest layer was above the PNW, between 44 °N and 48 °N. The simulated aerosol mixing layer height (Figure 3c) is slightly lower than that observed in the satellite data (Figure 3a,b), which might be attributable to biases in the plume rise algorithm in the CMAQ model system. We used the default Briggs plume rise algorithm [46] in the SMOKE emission module of the CMAQ system, which was originally developed for power plant stacks and was modified by converting the heat flux to a buoyancy flux for fire smoke plume. This scheme is more suitable for small fires than large wildfires because of its weak representation of microphysical processes affecting plume rise simulations [47]. Other plume rise parameterizations will be tested in the future study to improve the simulation of fire smoke vertical distribution. The CATS product also identified different aerosol types. Most of the aerosols in this region were categorized as smoke (category 7 in black; Figure 3b). Other aerosol types such as dust (category 3 in light yellow; Figure 3b) and sea salt/marine aerosols (category 1 in deep blue; Figure 3b) were present at the same time, but these aerosols mainly dominated other regions outside the model domain. 

#### 3.2.2. Data Fusion for Surface PM_2.5_ Concentrations

After gap filling for missing values in MAIAC AOD, we applied the same data fusion algorithms to optimize surface PM_2.5_ and smoke exposure estimates based on AirNow in situ observations and WRF-CMAQ model simulations, gap-filled AOD, and reanalysis-based meteorological variables as introduced in Section 2. Figure 4 shows the regional PM_2.5_ simulation performance of the raw WRF-CMAQ SENS system, which incorporated emission contributions from all major source sectors including both fire and non-fire emissions. 

The monthly averaged surface PM_2.5_ concentration field shows major hotspots in these mountain forests on both sides of the PNW, especially along the Western Cascades and the Bitterroot Range, which are consistent with the satellite imagery in Figure 1. These fire hotspots released large amounts of aerosols and tracer gases in forms of heavy smoke that enveloped almost the whole PNW (Figure 4a). Although the model results show good agreement, with high temporal correlations with AirNow ground monitoring data at most sites (Figure 4b), the negative biases in simulated PM_2.5_ surface concentrations are pervasive over the whole model domain (Figure 4c). Such underestimation in surface PM_2.5_ simulation may suggest systematic low bias in primary emission inventories and/or insufficient secondary aerosol formation in simulated fire plumes, both of which require further investigation and correction by pre- and post-processing measures. The site-specific RMSE results (Figure 4d) show a similar pattern with the regional PM_2.5_ concentrations (Figure 4a), manifesting high values in near-source mountain regions and low values in downstream regions after fire smoke transport and dispersion. This pattern can be explained by the calculation method of RMSE (see Appendix A), which is based on the absolute values of PM_2.5_ concentrations. Because of the large uncertainty in fire emission inventory, these PM_2.5_ concentrations that are heavily dominated by fire smoke in near-source regions may suffer a larger influence from bias in estimation of fire emission, contributing to larger RMSE values at near-source sites (Figure 4d). 

To solve the systematic low-bias problem in the WRF-CMAQ SENS experimental results, we applied the three data fusion methods to the raw model results and generated corresponding surface PM_2.5_ concentration fields for comparison with the AirNow ground monitoring data. We selected 16 ground sites (Figure 4a) with representative spatial and population coverage for the PNW and evaluated temporal variations of daily PM_2.5_ concentrations at these 16 sites during the fire smoke pollution episode. All sites experienced at least one non-attainment day, with daily PM_2.5_ concentrations exceeding the 24-hour threshold of 35 μg m^−3^ in the National Ambient Air Quality Standards throughout the study period (Figure 5). Some near-source sites, such as site-3 (Spokane, WA), site-9 (Springfield, OR), and site-10 (Bend, OR), experienced exceptional non-attainment days with highest pollution peaks beyond 200 μg m^−3^. Exposure to such heavy smoke pollution even over a brief period would cause severe health problems to both sensitive population groups, such as patients with pre-existing conditions (e.g., asthma and chronic obstructive pulmonary disease), and healthy people [4]. Although the raw WRF-CMAQ SENS results (Figure 5, blue dashed lines) well captured temporal evolution of surface PM_2.5_ with good temporal correlations with ground observations (Figure 5, red solid lines) at most sites, the low bias emerged again at all selected sites. The MLR results (Figure 5, green dashed lines) somewhat alleviated the low-bias problem but failed to increase temporal correlations for most sites. In comparison, the RF (Figure 5, golden dashed lines) and the GBM (Figure 5, purple dashed lines) methods greatly improved both the accuracy and correlation of modeling results at most ground sites. The RF method outperformed the GBM method, in slightly better agreement with observations.

The scatter plots in Figure 6 give a better presentation of comparisons between the daily modeling results and the AirNow in situ observations. As discussed above, the raw CMAQ SENS results without data integration show upward shifted distributions, with the data centroid above the 1:1 reference line (Figure 6a), suggesting low bias in the raw modeling results (Table 4). After model parameter tuning (Appendix A), all three data fusion algorithms improved the data distributions to different extent, with the best performance in the RF results in most statistical metrics (Figure 6c, Table 4). The regionally averaged PM_2.5_ concentration increased significantly after data integration, especially weighted by population density. The area-weighted regional average PM_2.5_ concentration increased by 90% in the ensemble mean of the three data fusion algorithms (MLR, RF, and GBM), and the population-weighted regional average PM_2.5_ concentration increased by 193% in the ensemble mean. Since the health impact assessment was based on population exposure to PM_2.5_ pollution, we mainly focus on the RF result that shows the best modeling performance of surface PM_2.5_ concentrations.

Before assessing regional health impacts with the surface PM_2.5_ exposure estimates, we evaluated PM_2.5_ chemical composition by comparing model-simulated aerosol species (the SENS results) with the in situ measurements from the IMPROVE network over the PNW region. Figure 7 shows the comparison results of seven chemical species at 12 IMPROVE sites. The pie sizes denote bulk PM_2.5_ concentrations and colored fractions denote the chemical composition, such as elemental carbon (EC), organic carbon (OC), sulfate (SO_4_), nitrate (NO_3_), ammonia (NH_4_), soil/dust, and sea salt. For all the selected sites, OC was the dominant aerosol species during the pollution episode, which is in good agreement with typical fire smoke chemical characteristics and independent in situ measurements [44]. This agreement is important to warrant the usage of the relative risk function in Equation (2) because the function was developed specifically for fire smoke exposure in Johnston et al. [1]. The model overestimates contributions from sea salt at many sites but underestimates soil aerosols at specific sites, such as site-5 (MOHO1 in Mount Hood, Oregon), site-9 (GLAC1 in Glacier National Park, Montana), and site-12 (JARB1 in Jarbidge, Nevada). Discussion of these chemical species is beyond the scope of this study, because they are from mostly non-fire sources with relatively small contributions to bulk PM concentrations. 

### 3.3. Regional Health Impact Assessment

Lastly, we conducted regional health impact assessment for the 2017 PNW fire smoke pollution episode following Johnston et al.’s method [1]. Figure 8a shows regional multiple-cause mortality at the county level over the PNW in August-December of 2017, and Figure 8b shows county-level deaths attributed to the PM_2.5_ pollution exposure based on the RF results. These regions that were heavily affected by PM_2.5_ pollution are consistent with the spatial distribution of major population centers over the PNW, which also have relatively large numbers of deaths. However, the mountain regions of the Western Cascades and the Bitterroot Range showed more significant health outcomes because of heavier fire smoke exposure in these two near-source areas than in the Puget Sound region, despite the much larger urban population and mortality. Fire smoke mostly blanketed the southwestern and northeastern corners of the PNW, with gradually decreasing PM_2.5_ concentrations from near-source regions to downwind regions (Figure 8c). Fire smoke severely undermined public health, with most fire smoke-related deaths occurring in Spokane County (12; 95% CI: 0, 28) of eastern Washington and Jackson County (11; 95% CI: 0, 26) of southwestern Oregon. Some distant downstream regions such as Salt Lake County (4; 95% CI: 0, 9) in Utah also witnessed moderate health effects of fire smoke. The population-weighted regional mean PM_2.5_ concentration based on the RF results is 13 μg m^−3^ (Table 4), to which fire sources contributed around 85% (11 μg m^−3^; Figure 8c) and non-fire sources contributed only 15% (2 μg m^−3^; Figure 8d). Though PM_2.5_ concentrations in the CMAQ_CTRL experiment (Figure 8d) were not corrected by observations, it’s noted that this estimate of the regional background PM_2.5_ concentration for the PNW is similar with a previous model-based study (2 μg m^−3^ averaged from 18 rural sites and 4 μg m^−3^ averaged from 36 urban/suburban sites with non-fire sources in the PNW) [14] and an observation-based study (3.3 μg m^−3^ averaged from seven background monitoring sites during non-fire season in the PNW) [48]. We also compared the time averaged CMAQ_CTRL results with the AirNow ground observations during a non-fire period (June–July, 2017) before the fire smoke episode (Appendix A). The comparison showed good agreement on a regional scale, suggesting the effective representativeness of the uncorrected CMAQ_CTRL results for non-fire sources. Based on the RF method, the estimated total number of regional deaths attributable to the 30-day PM_2.5_ exposure was 183 (95% CI: 0, 432), and fire smoke was the largest PM_2.5_ pollution source. This estimate varies little if we change the data integration method to the other algorithms. The uncertainty range is ~10% among the three data integration methods, with a similar estimate based on the GBM results (182; 95% CI: 0, 431) and a higher estimate based on the MLR results (202; 95% CI: 0, 477). The total PM_2.5_ attributable regional deaths would decrease significantly to 9 (5% CI: 0, 20) if without the fire events in the CTRL experiment, suggesting even larger contributions (95%) of fire emissions to regional health effects due to the piecewise feature in the exposure-response function (Equation (2)).

## 4. Discussion

In this study, we developed a two-step data integration approach based on a regional air quality modeling system and multiple space- and ground-based observations using three statistical algorithms. The modeling results after data integration showed significant improvement in terms of several modeling performance metrics, such as reduced bias and increased spatiotemporal correlations. Though we used the same tree-based data integration methods with these in Reid et al. [33], the improvement in our RF and GBM results was less significant than that in their study. These discrepancies may result from differences in model training datasets. They used much more predictor variables (29 variables in total) to train their models than we did in this study. Adding more predictor variables could increase the explainability of statistical models but sacrifice the model size with possible redundancy and collinearity problems. Several variable selection approaches such as stepwise or criterion-based procedures will be employed for further model selection and optimization in the operational system. After data integration, we applied the optimized PM_2.5_ exposure estimate to regional health impact assessment and evaluated adverse health outcomes of the pollution episode in terms of multiple-cause mortality. This application demonstrates the profound health impact of regional fire smoke across the PNW given the large increase in the PM_2.5_-related multiple-cause mortality. It’s noted that the health impact of fire smoke could be more serious if we consider additional PM_2.5_-related health outcomes and other air pollutants in fire smoke. In a fire smoke health impact study for British Columbia, Canada, that is adjacent to the PNW, researchers found 5% to 15% increased odds of respiratory physician visits and hospital admissions after smoke exposure over a 3-month study period in 2003 [49]. Though they did not find associations of cardiovascular outcomes with fire smoke, another health study conducted in Victoria, Australia, reported increased risk of acute coronary events, including out-of-hospital cardiac arrests and ischemic heart disease, during wildfire episodes [50]. In the USA, the total PM_2.5_-related and ozone-related deaths were estimated about 130,000 and 4700, respectively, based on the 2005 air quality levels from all emission sources [41]. Since wildfires are an important contributor to both particulate matters and tropospheric ozone production [51], it’s worthy to expand the research scope to include both PM_2.5_-related and ozone-related health effects during fire events. 

This study also highlights great potential of the ML-based data integration approach for not only health impact assessments of fire smoke exposure, but also operational air quality forecasting and reanalysis operations for early warning of fire smoke hazards. Previous studies suggested improved public health preparedness with integrated fire smoke forecast products [52] and huge economic benefit of forest-based interventions by reducing the health and economic burden of wildfires [53]. The interagency WFAQRP program led by the USFS would provide an effective working mechanism to achieve this goal with broad socioeconomic benefit for the USA. The nationwide deployment of ARAs supported by this program has been increasing throughout the past few years. These ARAs serve as a bridge between timely fire smoke information and the vulnerable population in fire-prone regions. With effective interventions through strengthened self and group protection, the adverse health effects and economic burden of fire smoke could be largely reduced.

To advance the research progress in related fields, we suggest three areas that deserve attention in future research. Firstly, continuous improvement of operational fire smoke forecasting and reanalysis systems and products at both regional and national scales is needed for long-term health impact assessments of fire smoke in the USA and elsewhere. This improvement would be enhanced from (a) application of newly available satellite observations, such as those from geostationary GOES-16/17 satellites with much higher temporal resolution, (b) advanced data-driven deep learning algorithms (e.g., Deep Convolution Neural Network), and (c) further development of fully coupled fire models with consideration of dynamic fire-atmosphere interactions across scales (e.g., WRF-SFIRE [54] and WRF-Fire [55]) as well as more comprehensive modeling capability of homogenous and heterogenous reactions for fire smoke aerosols. At global scales, one could apply continuously updating satellite-based global fire emission products [56] and newly developed process-based fire models [57] embedded in state-of-the-science earth system models for retrospective/predictive health and socioeconomic assessment of broad fire impacts.

Besides the spatial and temporal expansion of this integrated approach, we also emphasize the importance of aerosol chemical composition and its influence on short-term and long-term health effects in different population subgroups. Previous studies have reviewed aerosol composition-specific health effects based on oxidative stress and aerosol toxicity from different emission sources and burning phases [58,59]. Changes in aerosol speciation would result in distinctly different health outcomes regarding respiratory and cardiovascular diseases, morbidity, and mortality, highlighting the need for source-specific health impact assessment studies in the future. 

Lastly, reduced uncertainties in exposure-response functions are also expected by advancing cohort-based health studies regarding fire smoke exposure and health effects. Currently, large uncertainties exist in fire-related health studies, with knowledge gaps between fire smoke exposure and health outcomes. The uncertainty range induced by the relative risk function [1] used in this study is larger than 200% of the estimated values themselves. Such great uncertainty impedes accurate evaluation of consequent socioeconomic impacts of fire smoke hazards. More health research focusing on specific mortality causes, health outcomes, and vulnerable population groups, as suggested by previous reviews [4,25], would narrow down these knowledge gaps and uncertainties.

## 5. Conclusions

We applied a two-step data integration approach with regional air quality modeling results and satellite and ground observations for fire smoke health impact assessment. We estimated 183 deaths attributable to PM_2.5_ exposure during the 2017 fire smoke pollution episode (8/15–9/14/2017) over the Pacific Northwest. PM_2.5_ emissions from fire contributed 85% to the regional aerosol pollution, while non-fire emissions from anthropogenic and natural sources contributed 15%. The fire contributions to the consequent regional PM_2.5_ attributable multiple-cause mortality increased to 95% due to the piecewise feature in the exposure-response function. We suggest further improvement of regional and national fire smoke forecasting and reanalysis systems to reduce population exposure to fire smoke hazards and resulting public health risks.

## Figures and Tables

**Figure 1 ijerph-16-02137-f001:**
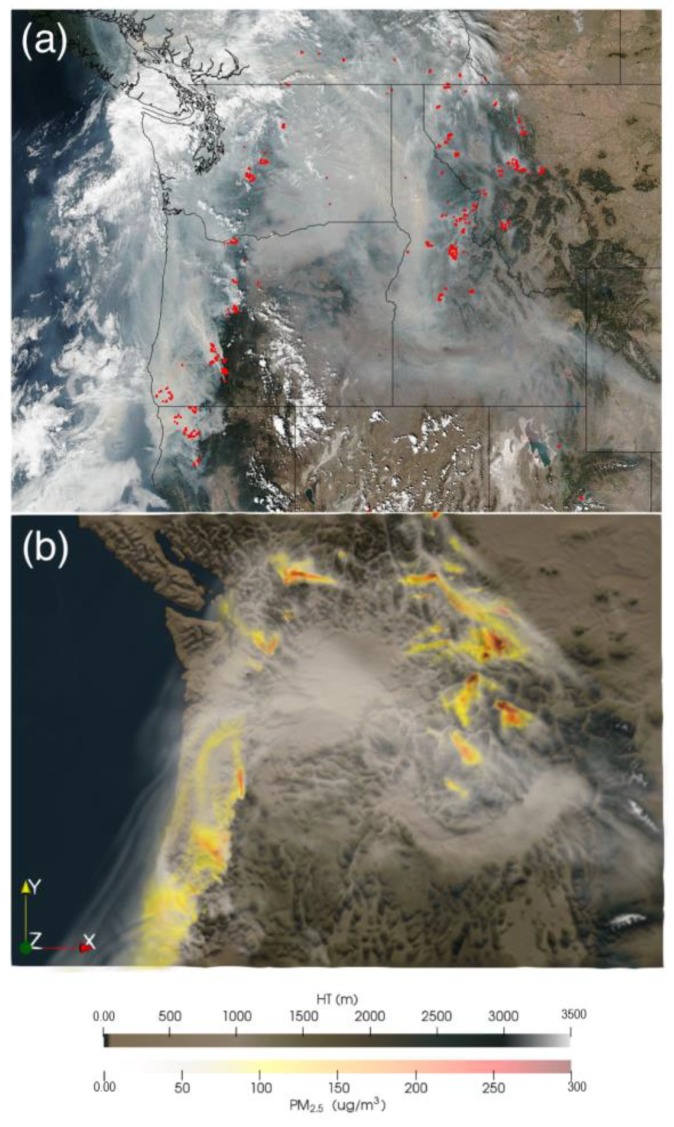
Fire hotspots and smoke over the PNW region on 9/5/2017. (**a**) satellite imagery detected by the NASA/NOAA Suomi NPP satellite at 20:36 UTC (13:36 PDT) (credit: Image by the NASA’s Land, Atmosphere Near real-time Capability for EOS (LANCE/EOSDIS) Rapid Response team). (**b**) WRF-CMAQ PM_2.5_ simulation based on the SENS experiment with all sources. PNW: Pacific Northwest. NASA: National Aeronautics and Space Administration. NOAA: National Oceanic and Atmospheric Administration. NPP: National Polar-orbiting Partnership. WRF: Weather Research and Forecasting model. CMAQ: Community Multiscale Air Quality model. PM_2.5_: Fine particulate matter. SENS: sensitivity experiment. UTC: Coordinated Universal Time; PDT: Pacific Daylight Time; EOS: Earth Observing System.

**Figure 2 ijerph-16-02137-f002:**
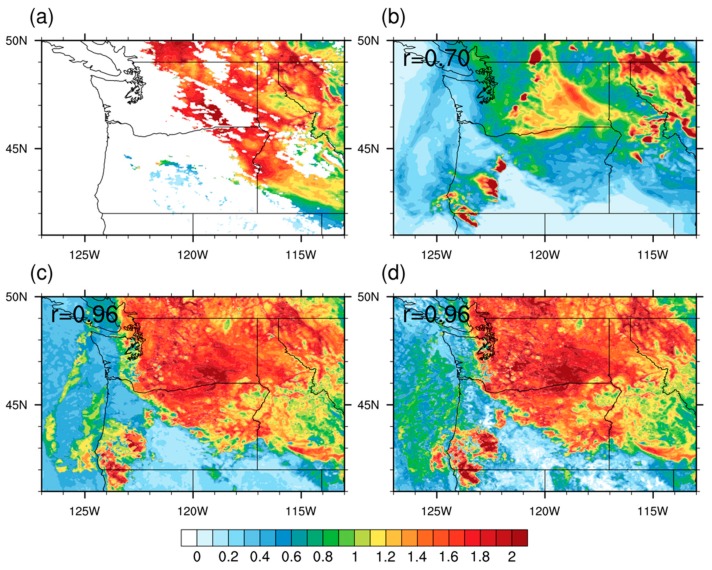
Comparison of the MAIAC AOD at 550 nm and model simulated AOD results at 20:15 UTC (13:15 PDT) on 9/7/2017. (**a**) the MAIAC AOD product onboard the MODIS Aqua satellite; (**b**) CMAQ simulated AOD; (**c**) bias-adjusted AOD by the RF method; (**d**) same as (**c**) but by the GBM method. The *r*-values on top-left corners of the subplots (b)–(d) denote spatial correlation coefficients between each model result and corresponding MAIAC AOD product. MAIAC: Multi-Angle Implementation of Atmospheric Correction. AOD: aerosol optical depth. MODIS: Moderate Resolution Imaging Spectroradiometer. CMAQ: Community Multiscale Air Quality model. RF: Random forest. GBM: Generalized boosting model.

**Figure 3 ijerph-16-02137-f003:**
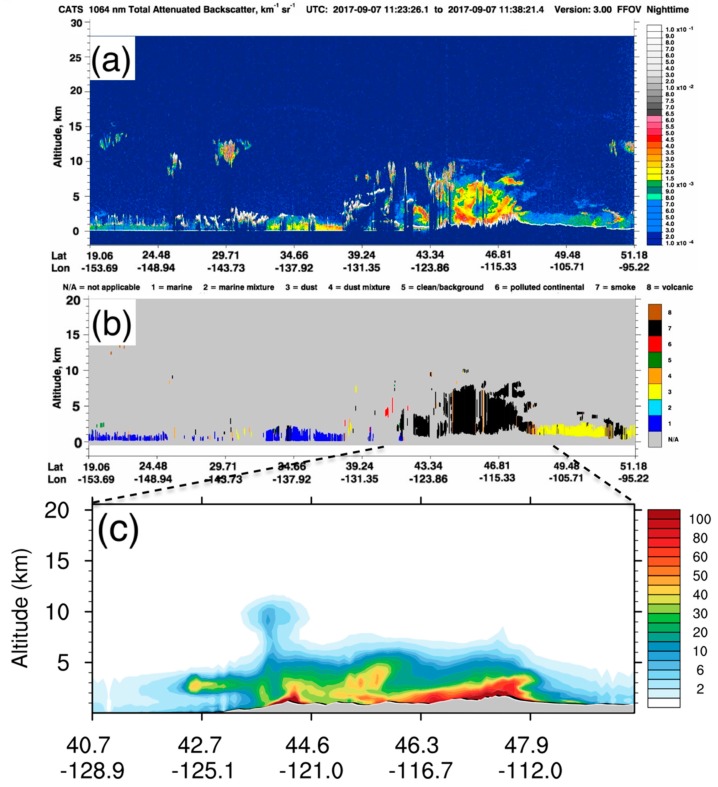
Comparisons of the aerosol vertical distribution at 11:30 UTC (04:30 PDT) on 9/7/2017 between the CATS satellite retrievals and the CMAQ simulation. (**a**) CATS total attenuated backscatter (unit: km^−1^ sr^−1^) at 1064 nm; (**b**) CATS aerosol types; (**c**) 2-D cross section of PM_2.5_ concentrations (unit: μg m^−^^3^) from the WRF-CMAQ SENS experiment along the CATS satellite track. Please note that the X-axis range in subplot (c) is different due to the smaller model domain. (credit: The CATS products in subplots (a) and (b) were produced and distributed by NASA Goddard Space Flight Center). CATS: Cloud-Aerosol Transport System. SENS: sensitivity experiment.

**Figure 4 ijerph-16-02137-f004:**
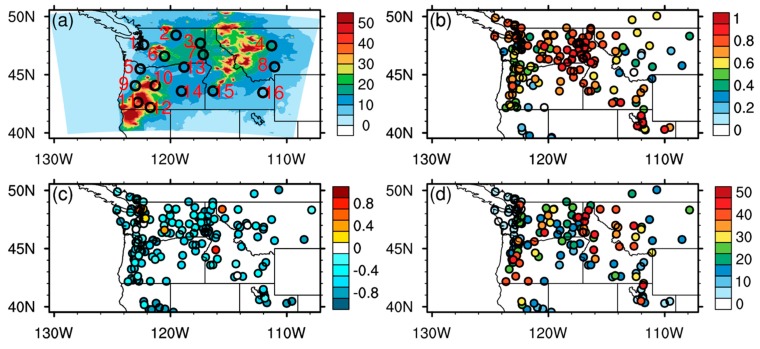
Statistical evaluation results of the raw WRF-CMAQ SENS simulated PM_2.5_ surface concentrations during 8/15–9/14/2017: (**a**) monthly averaged PM_2.5_ surface concentrations (unit: μg m^−3^). The black circles and red numbers denote 16 selected ground sites for the time series evaluation in Figure 5; (**b**) temporal correlations (unitless) between the SENS simulation and AirNow observations at each ground site; (**c**) fractional biases (unit: 100%) based on the SENS simulation and AirNow observations; (**d**) RMSE (unit: μg m^−3^) based on the SENS simulation and AirNow observations. RMSE: root mean squared error.

**Figure 5 ijerph-16-02137-f005:**
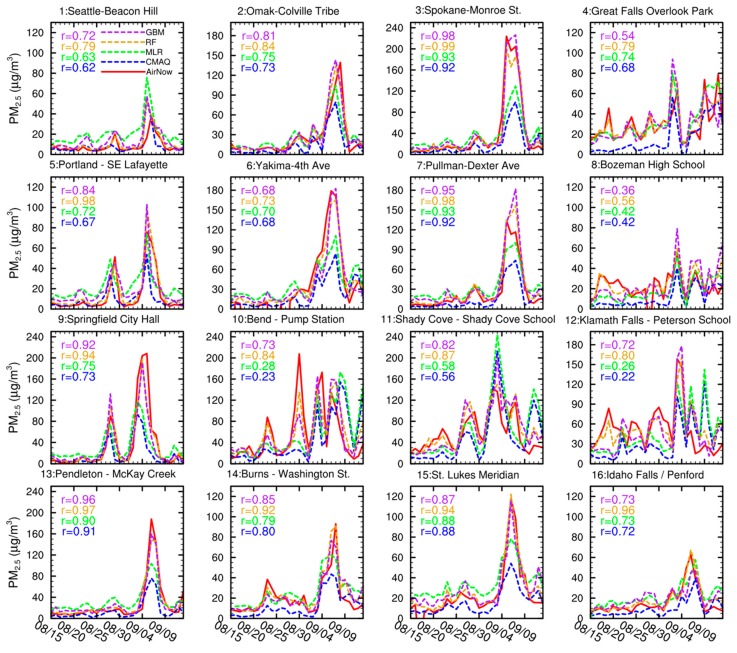
Time series of daily PM_2.5_ concentrations at the 16 selected AirNow ground sites in Figure 4a. The red solid lines denote AirNow ground measurements. The blue dashed lines denote CMAQ simulation results. The green dashed lines denote the MLR results. The golden dashed lines denote the RF results. The purple dashed lines denote the GBM results. The numbers on top-left corners of each subplot denote temporal correlations between these model results and observations. MLR: Multi-linear regression.

**Figure 6 ijerph-16-02137-f006:**
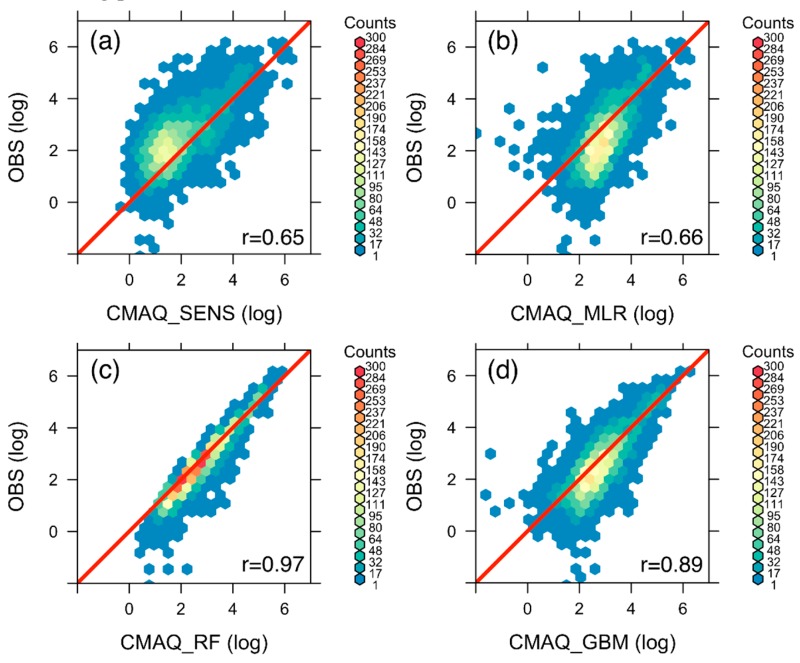
Comparisons of daily PM_2.5_ model simulation before and after data integration with AirNow ground observations during the fire smoke pollution episode. (**a**) a scatter plot of PM_2.5_ concentrations on a log scale based on the raw WRF-CMAQ SENS simulation and AirNow observations; (**b**) same as (**a**) but based on the MLR data integration method; (**c**) same as (**a**) but based on the RF method; (**d**) same as (**a**) but based on the GBM method. The color shading in all subplots denotes the PM_2.5_ data density in terms of sample counts.

**Figure 7 ijerph-16-02137-f007:**
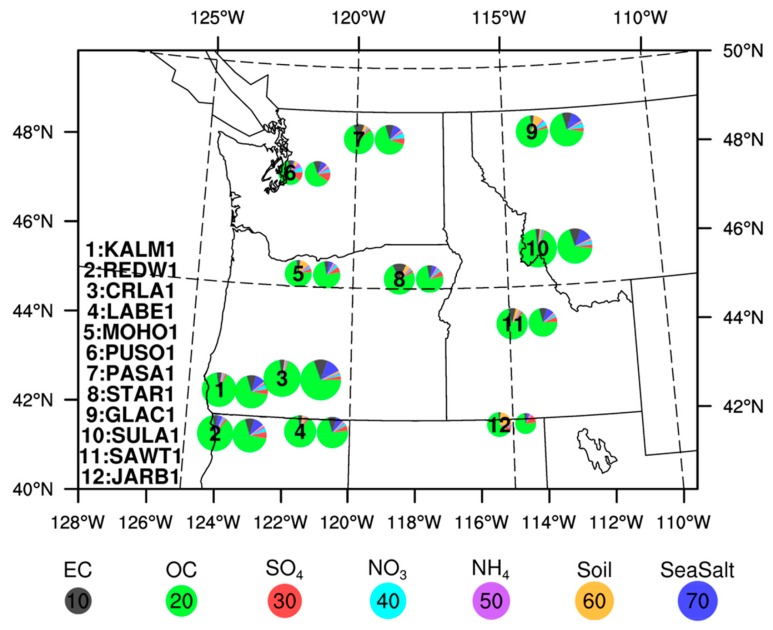
Comparison of PM_2.5_ concentrations and speciation between the IMPROVE ground measurement network (left pie charts with site numbers) and the WRF-CMAQ SENS model simulation (right pie charts without site numbers). The colors in each pie chart denote PM_2.5_ chemical compositions and the size of each pie chart denotes bulk PM_2.5_ concentrations as shown by the reference numbers in the legend.

**Figure 8 ijerph-16-02137-f008:**
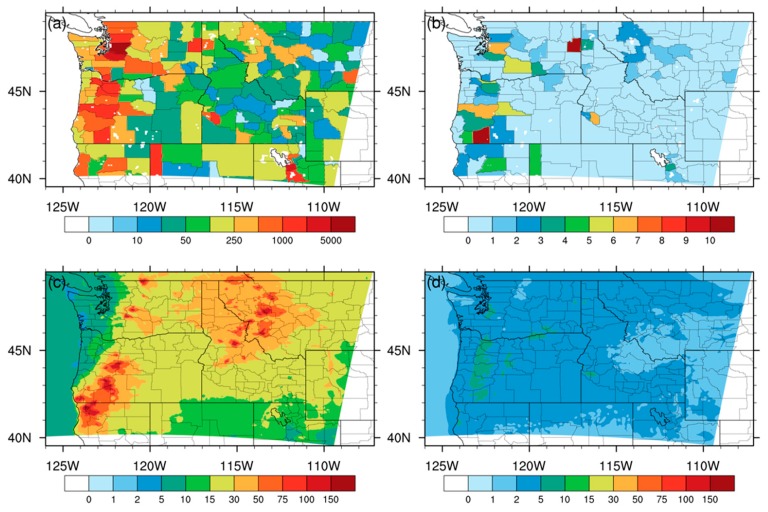
PM_2.5_ pollution exposure evaluation and attribution. (**a**) multiple-cause mortality (unit: ^#^) at the county level over the PNW in August-December of 2017; (**b**) county-level deaths (unit: ^#^) attributed to the PM_2.5_ exposure during the 2017 fire pollution episode; (**c**) PM_2.5_ concentrations (unit: μg m^−3^) contributed by fire sources (CMAQ_RF minus CMAQ_CTRL); (**d**) PM_2.5_ concentrations (unit: μg m^−3^) contributed by non-fire sources (CMAQ_CTRL).

**Table 1 ijerph-16-02137-t001:** The model simulation settings for the 2017 fire smoke case study over the PNW region.

Settings	CTRL	SENS
Period	08/13-09/14/2017 ^1^	08/13-09/14/2017 ^1^
Resolution	Horizontal: 4 km Vertical: 37 layers	Horizontal: 4 km Vertical: 37 layers
Meteorology	WRFv3.7 [27]	WRFv3.7 [27]
Chemistry	CMAQv5.2 [28] with cb05e51_ae6_aq	CMAQv5.2 [28] with cb05e51_ae6_aq
Fire emission	None	BlueSky [12]
Non-fire emissions ^2^	NEI2014 [32]	NEI2014 [32]
Initial/Boundary conditions	Prescribed concentrations	Prescribed concentrations

^1^ The first two-day model runs are for spin-up only and discarded for model evaluation. ^2^ non-fire emissions include both anthropogenic sources from power plants, industries, traffic, etc., and natural sources such as dust, sea salt, and biogenic emissions. PNW: Pacific Northwest; CTRL: control experiment; SENS: sensitivity experiment.

**Table 2 ijerph-16-02137-t002:** The observational datasets used for the model performance evaluation.

Metrics	Data	Resolution	Source
Horizontal distribution	MCD19A2 Version 6 MAIAC AOD [34]	Daily/1 km pixel size	NASA LP DAAC ^1^
Vertical distribution	CATS L1B v3.00 aerosol cross section [39]	Several times per day; Vertical: 60 m; Horizontal: 350 m	NASA GSFC ^2^
Temporal variation	AirNow PM_2.5_ surface concentrations	Hourly/in situ	The USA EPA ^3^
Chemical composition	IMPROVE aerosol speciation	Hourly/in situ	Inter-agencies ^4^

^1^https://lpdaac.usgs.gov/products/mcd19a2v006/. ^2^https://cats.gsfc.nasa.gov/. ^3^https://www.airnowtech.org/ index.cfm. ^4^http://vista.cira.colostate.edu/Improve/. MAIAC: Multi-Angle Implementation of Atmospheric Correction. AOD: Aerosol optical depth. NASA: National Aeronautics and Space Administration. LP DAAC: Land Processes Distributed Active Archive Center. CATS: Cloud-Aerosol Transport System. PM_2.5_: Fine particulate matter. GSFC: Goddard Space Flight Center. EPA: Environmental Protection Agency. IMPROVE: Interagency Monitoring of Protected Visual Environments.

**Table 3 ijerph-16-02137-t003:** The demographic data and relative risk function used for health impact assessment.

Data	Description	Source
Population	The USA Census Grids, 2010 [43]	NASA SEDAC ^1^
Mortality	Multiple cause of deaths in August–December of 2017	CDC WONDER ^2^
Relative risk function for multiple-cause mortality	0.11% (95% CI: 0, 0.26%) per 1 μg m^−^^3^ increase of surface PM_2.5_ concentration	Johnston et al. [1]

^1^http://sedac.ciesin.columbia.edu/data/set/usgrid-summary-file1-2010. ^2^https://wonder.cdc.gov/mcd.html. SEDAC: Socioeconomic Data and Applications Center. CDC: Centers for Disease Control and Prevention. WONDER: Wide-ranging ONline Data for Epidemiologic Research.

**Table 4 ijerph-16-02137-t004:** The modeling performance before and after data integration based on 10-fold CV.

Metrics	CMAQ_SENS	CMAQ_MLR	CMAQ_RF	CMAQ_GBM
Area-weighted regional average (μg m^−^^3^)	11.9	23.4	22.9	21.4
Population-weighted regional average (μg m^−^^3^)	4.5	15.9	12.8	10.9
MAE (μg m^−^^3^)	17.0	17.4	13.7	15.0
FB (%)	−44%	−1%	1%	−1%
R^2^ (unitless)	0.42	0.45	0.59	0.54
RMSE (μg m^−^^3^)	36.1	33.8	28.8	30.3

CV: cross-validation. MAE: Mean absolute error. FB: Fractional bias, R^2^: R-squared.

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
