# Peer review of "Machine Learning-Based Integration of High-Resolution Wildfire Smoke Simulations and Observations for Regional Health Impact Assessment"

_ijerph, 2019, doi:10.3390/ijerph16122137_

Round 1

Reviewer 1 Report

This study integrated numerical simulations and observations of regional fire events, and implemented three data fusion algorithms to reduce modeling bias in fine particulate matter and to optimize smoke exposure estimates. The manuscript was well-written. Methods and results were clearly introduced and explained. However, there were other studies investigating fire-induced mortality but the manuscript did not provide a comparison of the results in this study with previous studies. I recommend publication after adding the comparison and addressing a few other problems listed below.  

1.     Although cross-validation is a common statistical method, please explain 10-fold cross-validation for readers convenience in the manuscript.

2.     The manuscript used a random forest method and a generalized boosting model following Reid et al. [2015]. Although applied over different regions, please comment on the performance of these two methods in the PNW case in this study compared with those in the California case [Reid et al., 2015]. For example, do different data sets used in Reid et al. [2015] and this study influence RMSE or R2?

3.     Line 198: Minimum and maximum daily average surface PM2.5 concentrations of 5 and 200 ug m-3 were used to calculate morality attributable to PM2.5 exposure. Please explain why these two thresholds were selected and provide references.

4.     Figure 3: Please explain or comment on why aerosols mix higher in (a) and (b) than in (c).

5.     Figure 6: How were the 16 sites selected? Were they selected based on PM2.5 concentration level? Or were they IMPROVE sites?   

6.     Figure 7: Are statistical algorithms also applied to adjust the proportion of PM2.5 compositions?

7.     Figure 8: Please explain why high fire-induced mortality regions (b) do not match high fire PM2.5 concentration regions (c), especially in north Idaho and west Montana.

8.     Line 425: Please provide references for the 200% uncertainty induced by the relative risk function.

9.     Line 438: The manuscript suggested further improvement of fire forecasting. Does it imply using the statistical methods applied in this study to improve fire prediction? These methods all relied on observation data to adjust modeling results. However, there is no observation to use for prediction. This sentence is confusing.

Minor:

1.     Line 201: The sentence “???2.5 is the number of days between each PM2.5 concentration interval (i.e., each 1 μg m-3 increment between 5 μg m-3 and 200 μg m-3)” is confusing.

2.     Line 224: “Similar techniques will also be applied to an operational air quality forecasting system (https://tools.airfire.org) deployed at the Pacific Wildland Fire Sciences Laboratory of the U.S. Forest Service (USFS) to improve daily air quality forecasting skills as well as to reduce public exposure risks to fire smoke hazards through the interagency Wildland Fire Air Quality Response Program led by the USFS.” Is this an USFS plan for sure?

3.     Figure 3: The font size is too small.

4.     Line 422: The first sentence is incomplete.

Author Response

Thank you very much for your review and recommendation.

This study integrated numerical simulations and observations of regional fire events, and implemented three data fusion algorithms to reduce modeling bias in fine particulate matter and to optimize smoke exposure estimates. The manuscript was well-written. Methods and results were clearly introduced and explained. However, there were other studies investigating fire-induced mortality but the manuscript did not provide a comparison of the results in this study with previous studies. I recommend publication after adding the comparison and addressing a few other problems listed below.

Response: Thank you very much for your constructive comments and suggestions. We revised the manuscript accordingly to address your concerns and improved the presentation quality of our work. Specifically, we added more discussion in lines 559-566 and lines 569-582 to compare with previous fire smoke health studies. It’s noted that these studies were conducted for different regions and time periods, which provide qualitative rather than quantitative comparisons with our results. Please see below the responses in blue to your additional comments.

1. Although cross-validation is a common statistical method, please explain 10-fold cross-validation for readers convenience in the manuscript.

Response: Thank you for the suggestion. We added the explanation for 10-fold CV in lines 239-243 of the revised manuscript.

2. The manuscript used a random forest method and a generalized boosting model following Reid et al. [2015]. Although applied over different regions, please comment on the performance of these two methods in the PNW case in this study compared with those in the California case [Reid et al., 2015]. For example, do different data sets used in Reid et al. [2015] and this study influence RMSE or R2?

Response: Thank you. We compared our model performance with that in the 2008 California case study [Reid et al., 2015] in lines 559-566 as you suggested. Please note that Reid et al. [2015] tested 11 statistical algorithms and used much more predictor variables than what we did in our study (29 variables in Reid et al. [2015] vs. 6 variables for AOD gap-filling and 8 variables for PM2.5 optimization in this work). Some of the prediction variables are redundant with each other (e.g., multiple AOD products) and may shift the balance between model size and model performance. Though model performance is a key evaluation criterion, mode size is also an important factor in this study, especially considering the potential application of the data integration method in our future operational fire smoke forecast systems. Larger model size with more input variables would increase the difficulty for data collection and the probability of system failure. Our goal here is to reach a balance between model performance and robustness.

3. Line 198: Minimum and maximum daily average surface PM2.5 concentrations of 5 and 200 ug m-3 were used to calculate morality attributable to PM2.5 exposure. Please explain why these two

thresholds were selected and provide references.

Response: We followed the estimation method in Johnston et al. [2012] and used the same threshold for PM2.5 concentrations. In their principal analysis, a linear relative rate (RRSI) estimate of 0.11% (95% confidence interval: 0, 0.26%) per increase of 1 μg m-3 PM2.5 was used with minimum and maximum concentrations of 5 and 200 μg m-3. This RRSI was calculated using the average of values from studies reporting associations between all-cause mortality and short-term elevations of ambient PM10 [Morgan et al., 2010; Sastry 2002] and PM2.5 [Hanninen et al., 2009] during fire events. The PM10 concentrations were converted to PM2.5 by assuming 75% of all PM10 were PM2.5 to build the relation in RRSI. Though they did not explicitly explain why they capped the maximum PM2.5 concentration to 200 μg m-3, we found similar capping procedures in other fire smoke health impact studies. In Kollanus et al. [2017], they fixed the maximum modeled daily PM2.5 concentrations to 100 ug/m3 to reflect the flat shape of the exposureresponse relationship at high concentration levels. We added the explanation and reference in lines 302-305.

4. Figure 3: Please explain or comment on why aerosols mix higher in (a) and (b) than in (c).

Response: Thanks. We added comments about the aerosol mixing layer heights and the plume rise

algorithm in lines 395-402.

5. Figure 6: How were the 16 sites selected? Were they selected based on PM2.5 concentration level?

Or were they IMPROVE sites?

Response: The 16 AirNow sites were selected based on the spatial representativeness. We selected both near-source and distant ground sites in a variety of terrains to evaluate fire smoke short-range and longrange transport. We also considered the demographic information to evaluate fire smoke simulations for major cities in the PNW. We added the comment in lines 447-448.

6. Figure 7: Are statistical algorithms also applied to adjust the proportion of PM2.5 compositions?

Response: No. The data fusion algorithms were applied to adjust the total amount of PM2.5 concentrations rather than its chemical compositions. The mortality function in Eqn. (2) only requires the bulk concentrations of PM2.5 for fire smoke exposure estimates. However, the relative risk function used in this equation was developed specifically for fire smoke pollution, which has its own chemical speciation characteristics. Therefore, we compared the model simulated PM2.5 chemical composition with the measurements from the IMPROVE network before applying the mortality equation for health impact assessment.

7. Figure 8: Please explain why high fire-induced mortality regions (b) do not match high fire PM2.5

concentration regions (c), especially in north Idaho and west Montana.

Response: The fire-induced mortality depends on two spatiotemporally varying factors: surface PM2.5 concentration fields (affecting ?"#$.& and ??()(??-..) in Eqn. (2)) and multiple-cause mortality distributions at the county level (affecting M in Eqn. (2)). Though north Idaho and west Montana show high PM2.5 concentrations because of many active fires in this region, the total numbers of multiple-cause mortality in each county are relatively low due to the low population density in this region (see Fig. 8a). Therefore, there might be mismatch between PM2.5 and fire-induced mortality distributions. A similar example is King County in WA, where shows high total mortality (Fig. 8a) but moderate (Fig. 8b) fire smoke attributable mortality (Fig. 8b) because of the relatively low fire-related PM2.5 concentrations in this area (Fig. 8c).

8. Line 425: Please provide references for the 200% uncertainty induced by the relative risk function.

Response: The uncertainty range is based on the 95% CI of the relative risk function (Table 3) in Johnston et al. [2012]. The relative risk would increase by 0.11% per 1 μg m-3 increase of surface PM2.5 concentration, but the range of its 95% CI is from 0 to 0.26%, which is more than two times of its absolute value (0.11%). We added the reference in line 615.

9. Line 438: The manuscript suggested further improvement of fire forecasting. Does it imply using the statistical methods applied in this study to improve fire prediction? These methods all relied on observation data to adjust modeling results. However, there is no observation to use for prediction. This sentence is confusing.

Response: The data fusion methods used in this study can be used to improve initial conditions for operational fire smoke forecast. An accurate initial condition is important for short-term fire smoke forecast, especially when there are a lot of leftover fire smoke from previous days in the model domain. The idea here is similar with data assimilation that is widely used in numerical weather forecast operations. We added the explanation in lines 333-357.

Minor:

1. Line 201: The sentence “???2.5 is the number of days between each PM2.5 concentration interval (i.e., each 1 μg m-3 increment between 5 μg m-3 and 200 μg m-3)” is confusing.

Response: ?"#$.& is estimated by counting the number of days with daily PM2.5 at similar concentration levels between the minimum and maximum thresholds. For instance, it can be the total number of days with daily PM2.5 concentrations between 5 μg m-3 and 6 μg m-3, or between 6 μg m-3 and 7 μg m-3, etc. We chose 1 μg m-3 increment here because the relative risk linearly increases per 1 μg m-3 increase of surface PM2.5 concentrations (see Table 3). We revised the sentence in lines 305-307.

2. Line 224: “Similar techniques will also be applied to an operational air quality forecasting system (https://tools.airfire.org) deployed at the Pacific Wildland Fire Sciences Laboratory of the U.S. Forest Service (USFS) to improve daily air quality forecasting skills as well as to reduce public exposure risks to fire smoke hazards through the interagency Wildland Fire Air Quality Response Program led by the USFS.” Is this an USFS plan for sure?

Response: Yes. This is part of the working plan for the AirFire team at the Pacific Wildland Fire Science Lab of the USFS. It’s partly funded by the NASA Health and Air Quality Applied Sciences Team (HAQAST) project to improve fire smoke modeling skills and to provide better fire smoke forecast products to the public through the WFAQRP program. Dr. Susan O’Neill and Dr. Sim Larkin, the two coauthors of this paper, are the PIs of the NASA HAQAST project.

3. Figure 3: The font size is too small.

Response: Thank you. We updated the figure with better presentation quality.

4. Line 422: The first sentence is incomplete.

Response: Thank you. We revised the sentence accordingly.

References

Johnston FH, Henderson SB, Chen Y, Randerson JT, Marlier M, et al. 2012. Estimated Global Mortality Attributable to Smoke from Landscape Fires. Environ Health Perspect 120 (5): 695-701

Morgan G, Sheppeard V, Khalaj B, Ayyar A, Lincoln D, Jalaludin B, et al. 2010. The effects of bushfire smoke on daily mortality and hospital admissions in Sydney, Australia, 1994 to 2002. Epidemiology 21(1):47–55.

Sastry N. 2002. Forest fires, air pollution, and mortality in southeast Asia. Demography 39(1):1–23.

Hanninen OO, Salonen RO, Koistinen K, Lanki T, Barregard L, Jantunen M. 2009. Population exposure to fine particles and estimated excess mortality in Finland from an East European wildfire episode. J Expo Sci Environ Epidemiol 19(4):414–422.

Kollanus, V.; Prank, M.; Gens, A.; Soares, J.; Vira, J.; Kukkonen, J.; Sofiev, M.; Salonen, R.O.; Lanki, T. Mortality due to Vegetation Fire-Originated PM2.5 Exposure in Europe-Assessment for the Years 2005 and 2008. Environ Health Persp 2017, 125, 30-37, doi:10.1289/Ehp194

Reviewer 2 Report

General Comments:

The manuscript provides an in-depth evaluation of the impact of regional wildfires from August-September 2017 in the public health of the impacts U.S. Pacific North West.  The work integrates the use of numerical simulations of smoke transport and dispersion with observations, by applying a WRF model for air quality. High-resolution satellite corrections are applied for PM2.5 concentrations. The corresponding estimated mortality contributed by fire emissions is reported. Overall, this is a very well written manuscript and the work appears to be correctly performed to report an interesting public health case. Before publication, the authors should address the following specific comments in the final version of the manuscript.

Specific Comments:

1) Lines 77-81: Following these lines that introduce the impact of air pollution from fire emissions and the simulation of ozone, PM2.5, and tropospheric NO2, it would be really important to insert a few statements about the chemistry that organic molecules from wildfires undergo in the presence of these oxidizers. Noteworthy, there have been a series of recent papers that have evaluated the reactivity and lifetime of tracers for biomass burning that need to be discussed in statements immediately after the indicated lines. The manuscript should include here a brief summary of the following key seven papers dealing with the concepts for important biomass burning tracers:

a) EA Pillar et al., 2014. Catechol Oxidation by Ozone and Hydroxyl Radicals at the Air-Water Interface. Environ. Sci. Technol. 48, 14352-14360.

b) L Yu at al., 2016. Molecular transformations of phenolic SOA during photochemical aging in the aqueous phase: competition among oligomerization, functionalization, and fragmentation. Atmos. Chem. Phys. 16, 4511-4527.

c) EA Pillar et al., 2017. Oxidation of Substituted Catechols at the Air-Water Interface: Production of Carboxylic Acids, Quinones, and Polyphenols. Environ. Sci. Technol. 51, 4951-4959.

d) A Lavi et al., 2017. Characterization of Light-Absorbing Oligomers from Reactions of Phenolic Compounds and Fe(III). ACS Earth and Space Chemistry 1, 637-646.

e) ACO Magalhães et al., Density Functional Theory Calculation of the Absorption Properties of Brown Carbon Chromophores Generated by Catechol Heterogeneous Ozonolysis. ACS Earth and Space Chemistry 1, 353-360.

f) Smith JD et al., Phenolic carbonyls undergo rapid aqueous photodegradation to form low-volatility, light-absorbing products Atmospheric Environment. 126: 36-44.

g) J Sun et al., 2018. Mechanisms for ozone-initiated removal of biomass burning products from the atmosphere. Environ. Chem. 15, 83.

2) The internal labels/key for all the series in Figure 5 are not readable in a print out and need to be considerably improved.

3) Lines 404-429: These lines should be rewritten as a standard paragraph. The bullets/numbering to the left should be deleted.

4) Line 27: “10-Fold”.

Author Response

Thank you very much for your review and recommendation.

The manuscript provides an in-depth evaluation of the impact of regional wildfires from August-

September 2017 in the public health of the impacts U.S. Pacific North West. The work integrates the use of numerical simulations of smoke transport and dispersion with observations, by applying a WRF model for air quality. High-resolution satellite corrections are applied for PM2.5 concentrations. The corresponding estimated mortality contributed by fire emissions is reported. Overall, this is a very well written manuscript and the work appears to be correctly performed to report an interesting public health case. Before publication, the authors should address the following specific comments in the final version of the manuscript.

Response: Thank you very much for the constructive comments and recommendation. We revised the manuscript to address your concerns. Please see below the responses in blue to your specific comments.

Specific Comments:

1) Lines 77-81: Following these lines that introduce the impact of air pollution from fire emissions and the simulation of ozone, PM2.5, and tropospheric NO2, it would be really important to insert a few statements about the chemistry that organic molecules from wildfires undergo in the presence of these oxidizers. Noteworthy, there have been a series of recent papers that have evaluated the reactivity and lifetime of tracers for biomass burning that need to be discussed in statements immediately after the indicated lines. The manuscript should include here a brief summary of the following key seven papers dealing with the concepts for important biomass burning tracers:

a) EA Pillar et al., 2014. Catechol Oxidation by Ozone and Hydroxyl Radicals at the Air-Water Interface. Environ. Sci. Technol. 48, 14352-14360.

b) L Yu at al., 2016. Molecular transformations of phenolic SOA during photochemical aging in the

aqueous phase: competition among oligomerization, functionalization, and fragmentation. Atmos. Chem.Phys. 16, 4511-4527.

c) EA Pillar et al., 2017. Oxidation of Substituted Catechols at the Air-Water Interface: Production of Carboxylic Acids, Quinones, and Polyphenols. Environ. Sci. Technol. 51, 4951-4959.

d) A Lavi et al., 2017. Characterization of Light-Absorbing Oligomers from Reactions of Phenolic Compounds and Fe(III). ACS Earth and Space Chemistry 1, 637-646.

e) ACO Magalhães et al., Density Functional Theory Calculation of the Absorption Properties of Brown Carbon Chromophores Generated by Catechol Heterogeneous Ozonolysis. ACS Earth and Space Chemistry 1, 353-360.f) Smith JD et al., Phenolic carbonyls undergo rapid aqueous photodegradation to form low-volatility, lightabsorbing products Atmospheric Environment. 126: 36-44.

g) J Sun et al., 2018. Mechanisms for ozone-initiated removal of biomass burning products from the atmosphere. Environ. Chem. 15, 83.

Response: Thank you for the suggestion. The chemical composition and transformation in fire smoke is important to understand its reactivity, toxicity, and climate effects. We appreciate your recommendations of the key related references and added a brief summary in lines 84-88 of the revised manuscript. Please note that the homogenous and heterogenous reactions in fire smoke are very complicated and may not be very well represented in the current chemical mechanism of the CMAQ modeling system. We placed more emphasis on the total mass of PM2.5 for health impact assessment and compared fire smoke aerosol compositions briefly with observations from the IMPROVE network in Fig. 7. This comparison is mainly for the justification of the relative risk function being used in Eqn. (2) since it was developed specifically for fire smoke pollution. We also discussed the importance of chemical speciation for fire smoke health studies in the discussion section.

2) The internal labels/key for all the series in Figure 5 are not readable in a print out and need to be

considerably improved.

Response: Thank you. We updated the figure with better presentation quality.

3) Lines 404-429: These lines should be rewritten as a standard paragraph. The bullets/numbering to the left should be deleted.

Response: Thank you. We rewrote the paragraph as you suggested.

4) Line 27: “10-Fold”.

Response: Thank you. It’s revised.

Reviewer 3 Report

This is a well-conceived and robust research effort to assess mortality impacts from a wildfire event in the Pacific Northwest (PNW).  The authors also allude to a secondary goal of demonstrating the need for a “high-performance fire smoke forecasting and reanalysis system” to reduce health risks in fire-prone regions.  I recommend the manuscript for publication, but have the following major comments:

-        This second objective is not stated clearly or supported in the paper with a literature search and discussion, but a concluding statement is found in the abstract and conclusions sections.  Also, the idea is briefly mentioned in lines 401-402:  “It also highlights great potential of the ML-based data integration approach for not only health impact assessments of fire smoke exposure, but also operational air quality forecasting and reanalysis operations for early warning of fire smoke hazards.”

The authors don’t explain what forecasting products are currently available and how or why these are deficient.  They also don’t explain how measurements (clearly not available in the future) will be used to forecast fires and on what temporal period these forecasts should be provided, making it difficult to determine whether the statement is valid or not.  Reanalysis is more readily understandable, but a case is not made that complex post-processing of data are warranted.

-        The methodology for how fire emissions were allocated needs to be explicitly explained beyond describing that a “zero-out” run was completed.  I would expect that subtracting uncorrected simulated concentrations (CMAQ_CTRL) from corrected simulated concentrations (CMAQ_RF) would result in an over-estimation of exposure to fire-induced PM—especially given the composition differences noted in Figure 7. 

Other edits/comments:

Line 73:  Change to “…attributed to a prevalence of wildires…”

Lines 108-109:  “The CMAQ  model  (https://www.epa.gov/cmaq)  is  a  numerical  air  quality  model  developed  to  simulate emissions and atmospheric chemistry.”  While SMOKE is used to allocate emissions, I would not say CMAQ is used to simulate emissions.

Line 200:  Line 221 states that PM2.5 concentrations were up to 471 ug/m3, yet this sentence says the concentrations were capped at 200 ug/m3.  Please explain the rationale for capping the values at this number.

Line 306:  Change “studying” to “study”.

Figure 5:  Suggest adding an explanation of why results are poor for Site 1 – Seattle.

Lines 376-378:  Why is the population-weighted regional mean PM concentration meaningful?  This average is used as a reference for the allocation of fire emissions (i.e., 85% of this mean is attributed to fire sources).  It’s not clear why a population-weighted mean would be used to derive sources, or how source contributions were calculated.  Figure 8 has a brief mention that the fire-induced PM contribution was determined by CMAQ_RF minus CMAQ_CTRL.  This methodology, however, could produce an over-estimation of attribution to fire sources.  Since CMAQ_RF is “corrected” and CMAQ_CTRL is not, it seems you would use the daily grid ratio of CMAQ_SENS:CMAQ_CTRL.  This would allocate CMAQ SENS PM concentrations based on a percentage versus subtracting an uncorrected value from a corrected value.  Regardless, the methodology should be explicitly explained and supported. 

Author Response

Thank you very much for your review and recommendation.

This is a well-conceived and robust research effort to assess mortality impacts from a wildfire event in the Pacific Northwest (PNW). The authors also allude to a secondary goal of demonstrating the need for a “high-performance fire smoke forecasting and reanalysis system” to reduce health risks in fire-prone regions. I recommend the manuscript for publication, but have the following major comments:

Response: Thank you for the recommendation. Please see below the responses in blue to your

comments.

- This second objective is not stated clearly or supported in the paper with a literature search and discussion, but a concluding statement is found in the abstract and conclusions sections. Also, the idea is briefly mentioned in lines 401-402: “It also highlights great potential of the ML-based data integration approach for not only health impact assessments of fire smoke exposure, but also operational air quality forecasting and reanalysis operations for early warning of fire smoke hazards.”

The authors don’t explain what forecasting products are currently available and how or why these are deficient. They also don’t explain how measurements (clearly not available in the future) will be used to forecast fires and on what temporal period these forecasts should be provided, making it difficult to determine whether the statement is valid or not. Reanalysis is more readily understandable, but a case is not made that complex post-processing of data are warranted.

Response: Thank you for the comment. We revised the manuscript to address your concern here. We have two fire smoke modeling systems in operation by the AirFire team at the fire lab of the USFS, one based on the CMAQ model, and the other based on the HYSPLIT model. The CMAQ system can deal with leftover fire smoke from previous days while the HYSPLIT system cannot. The data fusion methods in this study can be used to improve initial conditions for CMAQ simulations by integrating observations into leftover smoke concentration fields. This is similar with the usage of reanalysis data in numerical weather forecast and is part of the working plan for the AirFire team at the fire lab. Please see detailed explanation in lines 333-357 of the revised manuscript.

- The methodology for how fire emissions were allocated needs to be explicitly explained beyond describing that a “zero-out” run was completed. I would expect that subtracting uncorrected simulated concentrations (CMAQ_CTRL) from corrected simulated concentrations (CMAQ_RF) would result in an over-estimation of exposure to fire-induced PM—especially given the composition differences noted in Figure 7.

Response: We only corrected PM2.5 bulk concentrations from all emission sources (CMAQ_SENS) for mortality estimates in Eqn. (2). No correction for chemical composition or PM2.5 concentrations from a hypothetical “fire-off” scenario (CMAQ_CTRL) has been done in the study. We compared aerosol chemical composition simulations and measurements in Fig. 7 to make sure the simulated chemical speciation agrees with the observations during the fire smoke episode (at least to some extent). This agreement is important to warrant the usage of the relative risk (RR) function in Eqn. (2) because this RR function was developed specifically for fire smoke pollution in Johnston et al. [2012]. We discussed the potential influence of different emission sources and burning stages on aerosol toxicity (and the RR function) in the discussion section. This is an active research area with lots of ongoing research progress as reviewed by Bates et al. [2019]. The composition-specific toxicity and health effects are beyond the scope of this study. Though the PM2.5 concentrations in the CMAQ_CTRL scenario were not corrected, it’s noted that the estimated PNW regional average concentration (2 μg m-3) from non-fire sources in Fig. 8d is similar with the simulated PM2.5 concentrations (2 μg m-3 for 18 rural sites in the IMPROVE network and 4 μg m-3 for 36 urban/suburban sites in the EPA-AQS network) without fire emissions in a previous model study [Chen et al., 2008] and an average concentration (3.3 μg m-3) of 7 background monitoring sites (4 IMPROVE sites and 3 urban background sites in the PNW) during non-fire season (April and May, 2004-2011) in an observation-based study [Timonen et al., 2013]. ]. We also compared the uncorrected CMAQ_CTRL results with that of the AirNow observations before the fire smoke episode in June-July, 2017, in Fig. S5 of the supplementary materials. We added these comparisons in lines 530-539 of the revised manuscript.

Other edits/comments:

Line 73: Change to “…attributed to a prevalence of wildires…”

Response: Thank you. It’s changed.

Lines 108-109: “The CMAQ model (https://www.epa.gov/cmaq) is a numerical air quality model developed to simulate emissions and atmospheric chemistry.” While SMOKE is used to allocate emissions, I would not say CMAQ is used to simulate emissions.

Response: Thank you. We revised the sentence accordingly.

Line 200: Line 221 states that PM2.5 concentrations were up to 471 ug/m3, yet this sentence says the concentrations were capped at 200 ug/m3. Please explain the rationale for capping the values at this number.

Response: We followed the mortality estimation method in Johnston et al. [2012] and used the same relative risk function with minimum/maximum concentration thresholds in their method. They did not explicitly explain why they capped the maximum PM2.5 concentration to this value, but we found similar capping procedures in other fire smoke health impact studies. In Kollanus et al. [2017], they fixed the modeled maximum daily PM2.5 concentrations to 100 ug/m3 to reflect the flat shape of the exposureresponse relationship at high concentration levels. We added the explanation in lines 208-217.

Line 306: Change “studying” to “study”.

Response: Thank you. It’s changed.

Figure 5: Suggest adding an explanation of why results are poor for Site 1 – Seattle.

Response: Thank you for the suggestion. The model results for the old site 1 at the 10th street of Seattle is poor because of many missing values in the observational data. These missing values reduced the sampling size for statistical evaluation and impeded the optimization by data integration. We replaced the old site with another one in Seattle (site 1: Beacon Hill) and updated Figs. 4/5.

Lines 376-378: Why is the population-weighted regional mean PM concentration meaningful? This average is used as a reference for the allocation of fire emissions (i.e., 85% of this mean is attributed to fire sources). It’s not clear why a population-weighted mean would be used to derive sources, or how source contributions were calculated. Figure 8 has a brief mention that the fire-induced PM contribution was determined by CMAQ_RF minus CMAQ_CTRL. This methodology, however, could produce an overestimation of attribution to fire sources. Since CMAQ_RF is “corrected” and CMAQ_CTRL is not, it seems you would use the daily grid ratio of CMAQ_SENS:CMAQ_CTRL. This would allocate CMAQ SENS PM concentrations based on a percentage versus subtracting an uncorrected value from a corrected value. Regardless, the methodology should be explicitly explained and supported.

Response: Thank you for the suggestion. The fire related mortality estimate depends on both PM2.5 concentrations and total mortality, which is related with population density (see Eqn. (2)). Therefore, population-weighted regional mean of PM2.5 concentrations is a better metric for population exposure than area-weighted regional mean. This metric is widely used in health assessment studies [e.g., Table 1 in Kollanus, 2017].

The usage of the daily grid ratio between CMAQ_SENS and CMAQ_CTRL for fire source contribution estimation has an assumption that the uncertainties in different emission sources (fire vs. non-fire) are similar or equivalent. However, this is not always the case since anthropogenic sources are relatively stable and well-studied and fire sources, especially large wildfires, are much more difficult to estimate given their rapid changes and great uncertainty ranges in fire emission model inputs (e.g., fuel loads, combustion efficiency, emission coefficients, etc.). As we pointed out in the above response, the CMAQ_CTRL results are in similar magnitude with previous modeling [Chen et al., 2008] and observation studies [Timonen et al., 2013]. We further compared the uncorrected CMAQ_CTRL results with that of the AirNow observations at 102 sites before the fire smoke episode in June-July, 2017, in a supplementary figure (Fig. S5). The comparison shows good agreement between the model results contributed by nonfire sources (averaged concentration at the model grid cells in which the 102 sites locate: 4.4 ug/m3) and the measurements during a non-fire period (regional averaged concentration of all 102 sites: 4.4 ug/m3), suggesting the effective representativeness of the CMAQ_CTRL results for non-fire emission contributions. The CMAQ_CTRL model simulations show a slight overestimation on a regional scale and the averaged fractional bias of all sites is 6%. Since the non-fire emissions were relatively stable without dramatic increases during the fire episode, we attributed the differences between the CMAQ_SENS results and the AirNow observations in Fig.4c to low biases in the fire emission estimates, which was then corrected by data fusion in CMAQ_RF. In this case, it’s proper to use the subtraction method and to attribute the discrepancies between CMAQ_RF and CMAQ_CTRL to fire sources. We revised the manuscript with more detailed explanation of this method in lines 530-539.

References

Johnston, F.H.; Henderson, S.B.; Chen, Y.; Randerson, J.T.; Marlier, M.; DeFries, R.S.; Kinney, P.; Bowman, D.M.J.S.; Brauer, M. Estimated Global Mortality Attributable to Smoke from Landscape Fires. Environ Health Persp 2012, 120, 695-701, doi:10.1289/ehp.1104422.

Bates, J.T.; Fang, T.; Verma, V.; Zeng, L.; Weber, R.J.; Tolbert, P.E.; Abrams, J.; Sarnat, S.E.; Klein, M.; Mulholland, J.A., et al. Review of acellular assays of ambient particulate matter oxidative potential: methods and relationships with composition, sources, and health effects. Environ Sci Technol 2019, DOI: 10.1021/acs.est.8b03430, doi:DOI: 10.1021/acs.est.8b03430.

Chen, J.; Vaughan, J.; Avise, J.; O'Neill, S.; Lamb, B. Enhancement and evaluation of the AIRPACT ozone and PM2.5 forecast system for the Pacific Northwest. J Geophys Res-Atmos 2008, 113, doi:Artn D1430510.1029/2007jd009554.

Timonen, H.; Wigder, N.; Jaffe, D. Influence of background particulate matter (PM) on urban air quality in the Pacific Northwest. J Environ Manage 2013, 129, 333-340, doi:10.1016/j.jenvman.2013.07.023.

Kollanus, V.; Prank, M.; Gens, A.; Soares, J.; Vira, J.; Kukkonen, J.; Sofiev, M.; Salonen, R.O.; Lanki, T. Mortality due to Vegetation Fire-Originated PM2.5 Exposure in Europe-Assessment for the Years 2005 and 2008. Environ Health Persp 2017, 125, 30-37, doi:10.1289/Ehp194.

Round 2

Reviewer 2 Report

Major comment:

The manuscript has not correctly addressed the previous major comments 1. Reviewer must reiterate the recommendation that following lines 77-81, the impact of air pollution from fire emissions and the simulation of ozone, PM2.5, and tropospheric NO2, needs to be highlighted with a few statements about the chemistry that organic molecules from wildfires undergo in the presence of these oxidizers.

In their revised manuscript the authors took the wording from this review to state that “It’s noted that some important atmospheric processes such as secondary aerosol formation and transformation by complex homogenous and heterogenous reactions [15,16] in fire smoke plumes are poorly represented in current chemical transport models. Continuous model development [17,18] are ongoing to improve the aerosol modeling capability for a better understanding of the role of aerosols in the climate system and human health.

Noteworthy, the recommendation had been to recognize a series of recent papers that have evaluated the reactivity and lifetime of tracers for biomass burning that need to be discussed in statements immediately after the indicated lines. The revision has only included the recommended references by Yu (item b below) and Smith  (items f below) but mistakenly avoided to summarize the other key 5 papers recommended (items a, c, d, e, and g below), that connect the importance of biomass burning tracers:

a) EA Pillar et al., 2014. Catechol Oxidation by Ozone and Hydroxyl Radicals at the Air-Water Interface. Environ. Sci. Technol. 48, 14352-14360.

b) L Yu at al., 2016. Molecular transformations of phenolic SOA during photochemical aging in the aqueous phase: competition among oligomerization, functionalization, and fragmentation. Atmos. Chem.Phys. 16, 4511-4527.

c) EA Pillar et al., 2017. Oxidation of Substituted Catechols at the Air-Water Interface: Production of Carboxylic Acids, Quinones, and Polyphenols. Environ. Sci. Technol. 51, 4951-4959.

d) A Lavi et al., 2017. Characterization of Light-Absorbing Oligomers from Reactions of Phenolic Compounds and Fe(III). ACS Earth and Space Chemistry 1, 637-646.

e) ACO Magalhães et al., Density Functional Theory Calculation of the Absorption Properties of Brown Carbon Chromophores Generated by Catechol Heterogeneous Ozonolysis. ACS Earth and Space Chemistry 1, 353-360.

f) Smith JD et al., Phenolic carbonyls undergo rapid aqueous photodegradation to form low-volatility, lightabsorbing products Atmospheric Environment. 126: 36-44.

g) J Sun et al., 2018. Mechanisms for ozone-initiated removal of biomass burning products from the atmosphere. Environ. Chem. 15, 83.

The manuscript should cite the above references and clearly state the limitations for modeling homogenous and heterogeneous reactions in fire smoke, which are very complicated and may not be very well represented in the current chemical mechanism of the CMAQ modeling system. In the conclusions sections the manuscript should mention the need to improve future models and studies by considering different homogeneous and heterogeneous processes.

Author Response

Thank you for the review.

Review report:

The manuscript has not correctly addressed the previous major comments 1. Reviewer must reiterate the recommendation that following lines 77-81, the impact of air pollution from fire emissions and the simulation of ozone, PM2.5, and tropospheric NO2, needs to be highlighted with a few statements about the chemistry that organic molecules from wildfires undergo in the presence of these oxidizers.

In their revised manuscript the authors took the wording from this review to state that “It’s noted that some important atmospheric processes such as secondary aerosol formation and transformation by complex homogenous and heterogenous reactions [15,16] in fire smoke plumes are poorly represented in current chemical transport models. Continuous model development [17,18] are ongoing to improve the aerosol modeling capability for a better understanding of the role of aerosols in the climate system and human health.

Noteworthy, the recommendation had been to recognize a series of recent papers that have evaluated the reactivity and lifetime of tracers for biomass burning that need to be discussed in statements immediately after the indicated lines. The revision has only included the recommended references by Yu (item b below) and Smith (items f below) but mistakenly avoided to summarize the other key 5 papers recommended (items a, c, d, e, and g below), that connect the importance of biomass burning tracers:

a) EA Pillar et al., 2014. Catechol Oxidation by Ozone and Hydroxyl Radicals at the Air-Water Interface. Environ. Sci. Technol. 48, 14352-14360.

b) L Yu at al., 2016. Molecular transformations of phenolic SOA during photochemical aging in the aqueous phase: competition among oligomerization, functionalization, and fragmentation. Atmos. Chem.Phys. 16, 4511-4527.

c) EA Pillar et al., 2017. Oxidation of Substituted Catechols at the Air-Water Interface: Production of Carboxylic Acids, Quinones, and Polyphenols. Environ. Sci. Technol. 51, 4951-4959.

d) A Lavi et al., 2017. Characterization of Light-Absorbing Oligomers from Reactions of Phenolic Compounds and Fe(III). ACS Earth and Space Chemistry 1, 637-646.

e) ACO Magalhães et al., Density Functional Theory Calculation of the Absorption Properties of Brown Carbon Chromophores Generated by Catechol Heterogeneous Ozonolysis. ACS Earth and Space Chemistry 1, 353-360.

f) Smith JD et al., Phenolic carbonyls undergo rapid aqueous photodegradation to form low-volatility, light absorbing products Atmospheric Environment. 126: 36-44.

g) J Sun et al., 2018. Mechanisms for ozone-initiated removal of biomass burning products from the atmosphere. Environ. Chem. 15, 83.

The manuscript should cite the above references and clearly state the limitations for modeling homogenous and heterogeneous reactions in fire smoke, which are very complicated and may not be very well represented in the current chemical mechanism of the CMAQ modeling system. In the conclusions sections the manuscript should mention the need to improve future models and studies by considering different homogeneous and heterogeneous processes.

Response to the reviewer:

Thank you for the quick review. In the original manuscript and the first-round revision, we briefly mentioned the limitation in current chemical transport models regarding secondary organic aerosols in fire plumes without detailed discussion of specific chemical species, reactions, and pathways mainly because we considered this work as a health assessment study rather than a chemical mechanism study. Considering the background of potential audiences of this paper, we tried to convey the information about SOA modeling limitations in plain language instead of complicated technical terms and jargon. However, we agree that these chemical pathways and reactions discussed in the recommended references are important for understanding the chemical evolution of secondary aerosols in fire plumes, especially these light-absorbing species with significant climate radiative effects. Though their influence on the health effects of ambient particles are not very clear, we want to bring to the attention of the potential audiences the complexity and great challenges of fire smoke modeling, especially when it involves SOA formation, transformation, and removal. Therefore, we reiterated the discussion about fire smoke modeling in lines 81-91 as follows:

“It’s noted that some important atmospheric processes such as secondary aerosol formation and transformation by complex homogenous and heterogenous reactions in fire smoke plumes are poorly represented in current chemical transport models (CTMs). Many recent studies have investigated chemical reactions of biomass burning products like catechol and phenolic compounds through oxidation at the air-water interface [15-17] or photodegradation in the aqueous phase [18-20]. These complex chemical pathways would affect secondary organic aerosol (SOA) formation and transformation in fire plumes and even removal of these biomass burning products initiated by ozone [21]. Though here we used a traditional organic aerosol treatment (CMAQ-AE6) that has limited modeling capability to reproduce these complex reactions, we want to point out that continuous model development [22,23] is ongoing to improve the aerosol modeling capability for a better understanding of the role of aerosols in the climate system and human health.”

We also revised the discussion section about the need of fire model development in lines 483-486:

“(c) further development of fully coupled fire models with consideration of dynamic fire-atmosphere interactions across scales (e.g., WRF-SFIRE [54] and WRF-Fire [55]) as well as more comprehensive modeling capability of homogenous and heterogenous reactions for fire smoke aerosols.”

We believe that the above revision has addressed your concern and we appreciate your comment to improve our work.

Round 3

Reviewer 2 Report

The revised manuscript has address all comments and is recommended for publication as is.